# Community Antibiotic Consumption in Cyprus for the Period 2015 to 2022

**DOI:** 10.3390/antibiotics13010052

**Published:** 2024-01-04

**Authors:** Eirini Mitsoura, Ioannis Kopsidas, Pambos Charalambous, Georgios Papazisis, Nikolaos Raikos, Zoi Dorothea Pana

**Affiliations:** 1Medical School, European University of Cyprus, 2404 Nicosia, Cyprus; mitsoura@auth.gr (E.M.);; 2Department of Clinical Pharmacology & Clinical Research Unit, School of Medicine, Aristotle University, 54124 Thessaloniki, Greece; papazisg@auth.gr; 3Center for Clinical Epidemiology and Outcomes Research (CLEO), 15451 Athens, Greece; 4IAS Sinopsis Ltd., 1071 Nicosia, Cyprus; pambos.charalambous@sinopsis.eu; 5Department of Forensic Science and Toxicology & Clinical Research Unit, School of Medicine Aristotle University, 54124 Thessaloniki, Greece; raikos@auth.gr

**Keywords:** antibiotic consumption, primary care, AWaRe, antimicrobial stewardship

## Abstract

Background: Cyprus currently reports to ESAC-Net the total consumption of antimicrobials, without distinguishing between hospital and community-based antibiotic use. As a result, these data can only provide generalized insights into antimicrobial trends in the country. Aim: This study is a first attempt to retrospectively analyze community antibiotic consumption in Cyprus for the period of 2015 to 2022. Material and Methods: Data on community antimicrobial consumption between 2015 and 2022 were extracted from Pharmatrack’s database. Orally administered dispensed antibiotics were categorized under the J01 group of the WHO Anatomical Therapeutic Chemical (ATC) classification and by the WHO’s AWaRe classification of antibiotics. Antibiotic consumption was calculated in both packages consumed and per 1000 inhabitants, overall, by year of consumption and districts. Results: During the period of 2015–2022, there was variability in the mean outpatient antibiotic consumption per 1000 inhabitants among the five districts in Cyprus. Community consumption increased by 38% throughout the study period. Additionally, a decrease of 3% in the consumption of WHO ‘Access’ antibiotics was observed, accompanied with a concurrent increase of 3% in the ‘Watch’ group. Specifically, in 2022 the WHO ‘Access’ group consumption in the Cypriot community was 48%, significantly lower than the WHO’s goal of 60% and the EU’s goal of 70% for ‘Access’ antibiotic consumption. Conclusions: Antibiotic consumption in the community of Cyprus between 2015 and 2022 demonstrated substantial variability among districts, with higher consumption in less populated areas. There was an increasing trend in community consumption over the years and a decreasing trend in the percentage of ‘Access’ antibiotics prescribed.

## 1. Introduction

Excessive use and misuse of antibiotics have contributed to the emergence of antimicrobial resistance (AMR), an alarming global public health concern. AMR is associated with an increase in the prevalence of difficult to treat infections, leading to prolonged hospitalization, increased healthcare costs, and increased morbidity and mortality rates [1,2]. According to the 2022 European Centre for Disease Prevention and Control (ECDC) Report, AMR causes more than 35.000 deaths each year as a direct consequence of infections that have become untreatable by antibiotics [3]. The COVID-19 pandemic disrupted normal healthcare services, including infection prevention and control measures, thereby increasing the risk of healthcare-associated infections (HAIs) [4]. Furthermore, the pandemic led to a reduction in surveillance activities and laboratory capabilities for detecting and monitoring AMR [5]. AMR efforts have been severely hampered, and surveillance and antimicrobial stewardship (AMS) programs have been marginalized, postponed, or halted. Furthermore, a significant increase in the use of antibiotics for the treatment of COVID-19 patients, both in hospitals and in the community, was reported [6]. This trend highlights the need to revive AMR prevention and control measures and efforts and re-introduce them into healthcare systems. Countries will need to intensify and enhance their efforts to combat AMR.

Cyprus developed a National Action Plan (NAP) in 2015, which was not implemented, as reported in the 2021 Tripartite AMR Country Self-Assessment Survey (TrACSS) [7]. The NAP’s objective of optimizing the use of antimicrobials was not met in 2020 and 2021, and limited adoption of the “AWaRe” classification of antibiotics in the National Essential Medicines List was observed [7]. The 2023 joint ECDC and WHO report highlighted that Cyprus exhibits one of the highest EU rates of antimicrobial resistance [1]. Notably, Cyprus is one of the four countries with *Escherichia coli* fluoroquinolone resistance over 50%, as well as one of the five countries to report *Enterococcus faecium* vancomycin resistance above 50% [8]. Additionally, Cyprus was ranked first and second in antimicrobial consumption among 29 European countries in 2020 and 2021, respectively [9].

Cyprus is currently reporting to ESAC-Net its total consumption of antimicrobials, without distinguishing between hospital- and community-sector antibiotic consumption. Consequently, ESAC-Net data can only provide a general overview of antimicrobial trends at a nationwide level. Stewardship targets for improvement in the community are not necessarily identical to the ones for hospitals. As the country emerges from the pandemic, there is a significant opportunity for Cyprus’s healthcare system to foster collaboration with policymakers, the public, and the media to raise awareness of antibiotic consumption and AMR data, and to support targeted future public health decisions. The present study is a first attempt to retrospectively analyze community antibiotic consumption in Cyprus and evaluate changes over time both before and during the COVID-19 pandemic.

## 2. Results

### 2.1. Total Community Antibiotic Consumption (per District) in Cyprus

The total community antibiotic consumption in Cyprus for the study period (2015 to 2022) was 5,696,045 packages. The antibiotic consumption by district, the overall consumption, and the consumption per 1000 inhabitants during the study time are shown in Table 1. Among the five regional districts, Famagusta, the district with the lowest reported population, presented the highest mean antibiotic consumption (969 packages/1000 inhabitants). Nicosia, the district with the largest reported population, presented the lowest antibiotic consumption.

### 2.2. Total Community Antibiotic Consumption over the Study Period (2015–2022) in Cyprus

Antibiotic consumption per 1000 inhabitants varied across the five districts of Cyprus over the last 7 years (Table 2). As shown in Table 2, in 2015, the antibiotic consumption ranged from 577 to 833 packages in Nicosia and Paphos, respectively. By 2022, consumption had increased to 859 and 1132 packages per 1000 inhabitants in Larnaca and Famagusta, respectively. Throughout the study period, community antibiotic use in Cyprus increased in all districts. In particular, overall community antibiotic consumption rose by 38.1% during the last 7 years (from 691 in 2015 to 954 in 2022). None of the five districts exhibited a significant decline or maintained the same consumption level over the study period. Nicosia recorded the highest increase in consumption over the last 7 years (58.8%), while Larnaca reported the lowest consumption increase (19.8%). Figure 1 graphically illustrates the gradual rise in consumption by district from 2015–2022, with slight fluctuations over the years. Starting from 2018, antibiotic consumption trended upward across all districts, continuing to rise during the COVID-19 pandemic period beginning 3 January 2020 [10]. Famagusta demonstrated the highest peak in 2020 with 1158 packages per 1000 inhabitants. All other districts demonstrated their highest consumption peaks in 2022, as shown also in Table 2.

### 2.3. Community Antibiotic Consumption Using ATC Classification (ATC Group J01) in Cyprus

Community antibiotic consumption in Cyprus was classified under the ATC classification (ATC group J01) for the study period from 2015 to 2022 (Figure 2). The J01 sub-groups whose consumption per 1000 inhabitants presented an increase within the community during the study period included the following: sulfonamides and trimethoprim (J01E); macrolides, lincosamides, and streptogramins (J01F); and quinolones (J01M). Consequently, J01E, J01F, and J01M demonstrated the biggest increases of 125.0%, 93.2%, and 49.1%, respectively. The changes in antibiotic consumption over the study period of 2015–2022, using the ATC classification, are shown in Table 3

### 2.4. Community Antibiotic Consumption Using WHO Aware Classification in Cyprus

Community antibiotic consumption in Cyprus was classified under the WHO AWaRe classification for the study period, revealing an almost consistent 50% consumption of ‘Access’ antibiotics, ranging from 51% to 50% between 2015 and 2021. However, in 2022, a decline in the outpatient consumption of ‘Access’ group antibiotics was observed (48%); more specifically, there was a decline of 5.02% compared to the previous year (Figure 3). Moreover, an increase in the consumption of antibiotics classified under the ‘Watch’ group was observed during the same year (5.80%).

The changes in antibiotic consumption over the study period in each district, using the AWaRe classification, are shown in Table 4. In 2022, compared to 2015, the percentage of the ‘Watch’ group rose by 3% while the ‘Access’ group had equal losses. At the district level, Nicosia showed an increase of 2% in the ‘Watch’ group, Limassol an increase of 3%, and Larnaca an increase of 2%. The most substantial changes over the study period were observed in Paphos and Famagusta, with ‘Watch’ antibiotic increases of 5% and 6%, respectively. Both Paphos and Famagusta, reporting the lowest initial rates of 45% for the ‘Watch’ category in 2015, progressed to reach 50%, matching the other districts. Larnaca is the sole district that maintained an ‘Access’ group proportion above 50%, whereas all other districts dropped below this threshold by the end of 2022.

## 3. Material and Methods

### 3.1. Study Area

We examined the consumption of antibiotics by analyzing the sales data from retail pharmacies across all five districts in Cyprus (Larnaca, Famagusta, Nicosia, Paphos, and Limassol) for the period from 2015 to 2022.

### 3.2. Data Collection and Analysis

#### 3.2.1. Antibiotic Consumption Data from Cypriot Pharmacies’ Database

Data on antimicrobial consumption in the community between 2015 and 2022 were extracted from Pharmatrack’s database. Pharmatrack collects data on medications dispensed from 256 pharmacies throughout Cyprus, representing over 50% of the total pharmacies across the country. Each pharmacy provides monthly sales data, which are then extrapolated to represent the entirety of Cyprus’s 547 pharmacies during the study period. The sample of pharmacies is stratified into homogeneous groups (large, medium, small), taking into consideration the region allocation and the pharmacy size, with large pharmacies defined as those with monthly turnover exceeding EUR 65,000 per month, small pharmacies defined as those with turnover below EUR 40,000, and medium-sized pharmacies falling in-between.

The sample of pharmacies is designed using a stratified, disproportional sampling technique, ensuring an appropriate sample allocation with the specified strata. This approach prioritizes pharmacies with the highest turnover, maximizing the likelihood of capturing data from pharmacies with the greatest influence on the market. Pharmacies included in the final sample are randomly selected, minimizing bias in the sample and securing accurate market estimates. Data from all items sold by pharmacies are extrapolated to provide market estimates, which are then consolidated into representative reporting for each category.

Data collection occurs on a monthly basis and includes information on dispensed drugs, the number and type of packages, and drug formulations. Drug consumption is measured and evaluated using the Tableau data software (www.tableau.com (accessed on 30 November 2023)). A statistical error of 4% was set for calculation purposes [11], in order for the sample design to meet the standard error requirements for the regional market breakdowns and the extrapolation to the whole country.

#### 3.2.2. Antibiotic ATC Classification

For the purposes of this study only data on orally administered antibiotic dispensing for drugs classified under the J01 group in the WHO Anatomical Therapeutic Chemical (ATC) classification were included [12].

#### 3.2.3. Antibiotic AWaRe Classification

We analyzed available data from 2015 to 2022, using the WHO’s AWaRe classification of antibiotics, a tool developed by the WHO to promote prudent antibiotic use at all levels of healthcare: local, national, and global [13]. Antibiotics were divided into 3 categories—‘Access’, ‘Watch’, and ‘Reserve’—based on their potential to contribute to AMR and the need to use them judiciously. This study investigated the consumption of antibiotics classified as ‘Access’ group antibiotics in the Cypriot community sector from 2015 to 2022. The findings were compared to the World Health Organization’s (WHO) target of at least 60% of total antibiotic consumption being from ‘Access’ group antibiotics [13].

#### 3.2.4. Antibiotic Consumption Indicators

Antibiotic consumption was calculated in packages consumed overall by regional unit and by year of consumption, by districts, and per 1000 inhabitants. For estimating consumption per 1000 habitants, we used freely available population data from the latest 2021 official census by the Statistical Service of Cyprus [14]. The previously available census was in 2011, and therefore calculations for all years of the study period were based on the 2021 census.

### 3.3. Statistical Analysis

The statistical analysis was conducted using the IBM SPSS (version 27) software. The antibiotic consumption was estimated from the sales volume of antibiotics sold in retail pharmacies in all five regions/districts in Cyprus (Larnaca, Famagusta, Nicosia, Paphos, and Limassol). Analysis of variance, in the form of one-way ANOVA, was employed to compare antibiotic consumption variances (in packages) across the five districts of Cyprus throughout the study period (2015–2022). Normal distribution was assessed using the Kolmogorov–Smirnov and Shapiro–Wilk tests.

## 4. Discussion

To our knowledge, this is the first study to analyze the community consumption of systemic antibiotics in Cyprus. According to the study results, for the period of 2015–2022, variability in the mean outpatient antibiotic consumption per 1000 inhabitants was observed among the five districts in Cyprus. The highest consumption was recorded in the districts of Famagusta and Paphos, the districts with the two smallest populations. Consumption in the community rose by 38% during the whole study period, from 691 packages of antibiotics consumed per 1000 Cypriots in 2015 to 954 packages in 2022. Additionally, it is noteworthy that over the study period a decrease of 3% of WHO ‘Access’ antibiotic consumption was observed, with a concurrent increase of 3% of the WHO ‘Watch’ group. Specifically, in 2022 the WHO ‘Access’ group consumption in the Cypriot community was 48%, significantly lower than the WHO’s goal of 60% and EU’s goal of 70% consumption of ‘Access’ antibiotics [13,15].

The WHO Expert Committee established the AWaRe classification of antibiotics in 2017, a valuable tool to support antibiotic stewardship efforts at local, national, and global levels [13]. This tool facilitates the oversight of antibiotic use, setting goals and supervising the results of stewardship strategies aiming at improving antibiotic use and mitigating antimicrobial resistance [16]. Antibiotics classified in the ‘Watch’ category have a higher potential for antimicrobial resistance selection and are more frequently administered in hospitals. To prevent overuse, their consumption should be closely monitored [17].

The World Health Organization’s 13th General Programme of Work 2019–2023 set a target of at least 60% of overall antibiotic usage falling within the ‘Access’ category [13]. In 2022, we estimated that Cyprus’s community consumption of ‘Access’ group antibiotics was at 48%, exhibiting a decline from 51% in 2015 overall, which is below the WHO target. While direct conclusions cannot be drawn, Table 2 reveals that the districts of Paphos and Famagusta exhibit increased antibiotic consumption rates within the ‘Watch’ group, ranging from 5% to 6% relatively. Consequently, it is reasonable to postulate that the 3% reduction in the ‘Access’ group among Cyprus’s overall antibiotic consumption in 2022, as depicted in Figure 3, can likely be attributed to the influence of these two districts.

ESAC-Net’s “Antimicrobial consumption dashboard” includes 2022 data on consumption [18], indicating that Cyprus (the third smallest EU country) reports 55.4% ‘Access’ combined consumption in the community and hospital sector [18]. This figure is comparable to those of other small countries, such as Malta’s 55%, and slightly lower than those of Croatia, Slovenia, and Luxembourg, which are 60.1%, 61.7%, and 60.8%, respectively. The community sectors of these countries also exhibit similar, albeit slightly higher, ‘Access’ percentages (55.40%, 61.7%, 62.9%, and 61.8%, respectively). However, as our data suggest, this is not the case for Cyprus, as consumption in the community is 48% for the ‘Access’ group. When compared to the community consumption of the other nine smallest countries in the EU, Cyprus appears to have the second lowest percentage of ‘Access’ group antibiotic consumption in the community sector. This highlights the significance of having granular data of consumption in the community and the hospital sector separately. Among the 27 European countries reporting data, our 48% ‘Access’ group finding would position Cyprus as the fourth lowest in ‘Access’ group consumption. Of note, further larger nationwide prospectively conducted studies are needed, as the antibiotic consumption indicators in the present study are defined in terms of packages dispensed, whereas the relevant ECDC data are reported in defined daily doses (DDDs).

Globally, low-middle-income countries (LMICs) appear to be ahead in achieving and exceeding the WHO target of 60% ‘Access’ consumption. Compared to Latin American antibiotic consumption, Cyprus and other European countries lag behind in reaching the 60% ‘Access’ goal. Argentina, Chile, Colombia, Costa Rica, and Peru have reportedly achieved or surpassed this goal, while Paraguay has not [19]. In Africa, high rates of ‘Access’ prescribing were reported in Ghana (60%), Uganda (60%), Zambia (58%), and Tanzania (59%) in a 2019 point prevalence study (PPS), which however did not look specifically in community settings [20]. A 2023 review by Chigome et al. in the community sector in Africa concluded that the majority of antibiotics prescribed were from the ‘Access’ group [21]. Looking in the Southeast Asia region, a study in Vietnam reports that 59% of sales were in the Access category, but these are not national data [22].

According to our results, consumption remained relatively stable from 2015 to 2018, a period before the COVID-19 outbreak. However, there was a consistent increase from 2020 to 2022, a timeframe that includes both the pre-COVID-19 and COVID-19 pandemic periods (Appendix A). This increase in outpatient antibiotic consumption during the pandemic might be due to the challenges of implementing antibiotic stewardship programs during the pandemic crisis, coupled with an increased antibiotic use for treating COVID-19 patients [23]. Nonetheless, it is important to note that our study was not designed to confirm such associations.

This study has a few limitations. Firstly, we did not include all of the pharmacies in Cyprus; approximately 50% were covered. However, extrapolation methods were utilized to estimate representative and unbiased sales data and package numbers for the whole country. Secondly, an analysis of packages dispensed per 1000 inhabitants was provided, which differs from the standard metric used in most of the ECDC’s reports, which use DDDs. This poses few issues when comparing data, but we believe that our results, being the first collection of community sales/consumption data, have value in understanding the community antibiotic consumption trends in Cyprus. Moreover, one person can receive more than one package to cover the duration and dose needs of an antibiotic course. This is, however, also a problem when calculating consumption using DDDs [24]. Prescriptions could be a better metric for measuring consumption in a community, provided that antibiotics cannot be procured over the counter, without a prescription [25]. Granular, anonymized, patient-level community prescription data could offer more insights into the reasons for which antibiotics are dispensed and more importantly regarding their appropriateness. Population censuses are conducted every 10 years. Prior to this study there were censuses in 2011 and 2021, with total populations of 856,960 and 904,700, respectively. Opting to use the 2021 population as the denominator for all years might result in an underestimation of packages sold per 1000 inhabitants in the study’s early years. Conversely, using the smaller 2011 population could lead to an overestimation of sales per 1000 inhabitants. We chose to use the 2021 population, as it would provide a more accurate representation for the last years of our study, which are closer to the situation in the present and can also be important for forthcoming public health policy.

As we are presenting community-level data for Cyprus for the first time, we believe that despite these limitations, our study contributes to a better understanding of community antibiotic consumption trends in the country over time. This could be useful when forging public health policies and future decisions.

## 5. Conclusions

Antibiotic consumption between 2015 and 2022 in the community of Cyprus showed variability across districts, with higher use in less populated areas and an overall increasing trend over time. Despite a decreasing trend in the use of ‘Access’ antibiotics, it still falls short of WHO goals. Data on antimicrobial consumption and stewardship are crucial in combating antimicrobial resistance, a significant issue in Cyprus. Systematic data collection is the first step towards strategic interventions in stewardship practices to address this challenge.

A contextualized roadmap against AMR is urgently needed, which will support the Cypriot community in developing tailored, meaningful, and sustainable solutions to antimicrobial overuse. Regardless of the importance of establishing and implementing national antimicrobial stewardship (AMS) policies and healthcare professional training strategies, the efficacy of antimicrobials will only be protected in the future if they are used appropriately as part of a more holistic approach across the One Health spectrum. Long-term behavior changes via public awareness-raising, but most importantly via systematic, continuous, and meaningful community engagement, is needed to efficiently combat AMR.

## Figures and Tables

**Figure 1 antibiotics-13-00052-f001:**
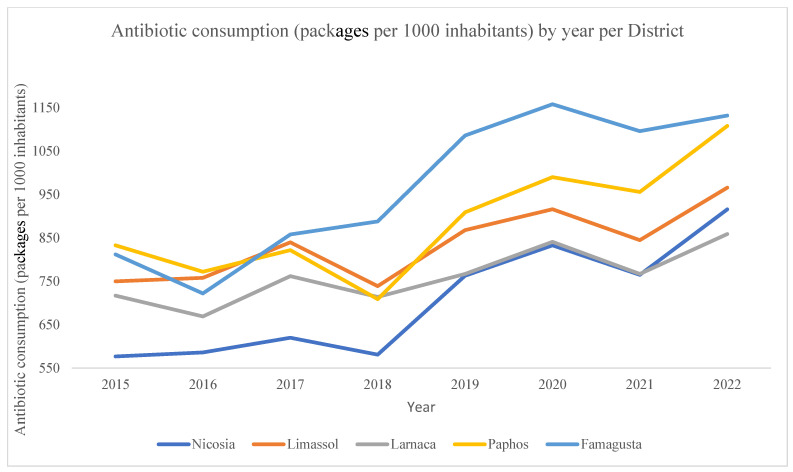
Antibiotic community consumption (packages dispensed) per 1000 inhabitants, yearly, by district, Cyprus, 2015–2022.

**Figure 2 antibiotics-13-00052-f002:**
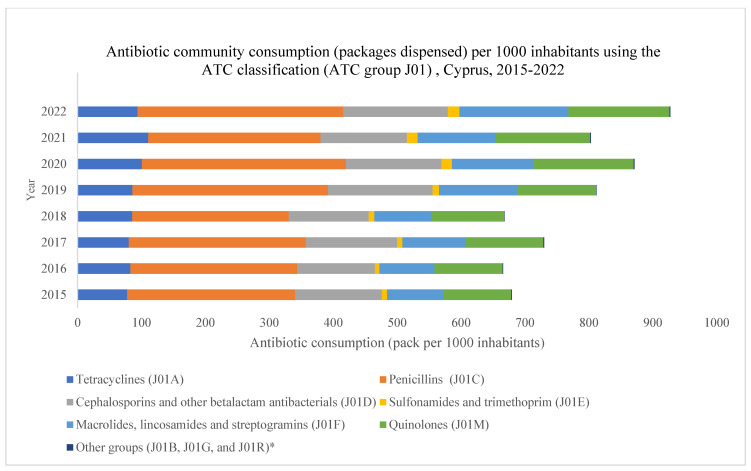
Antibiotic community consumption (packages dispensed) per 1000 inhabitants using the ATC classification (ATC group J01), Cyprus, 2015–2022.

**Figure 3 antibiotics-13-00052-f003:**
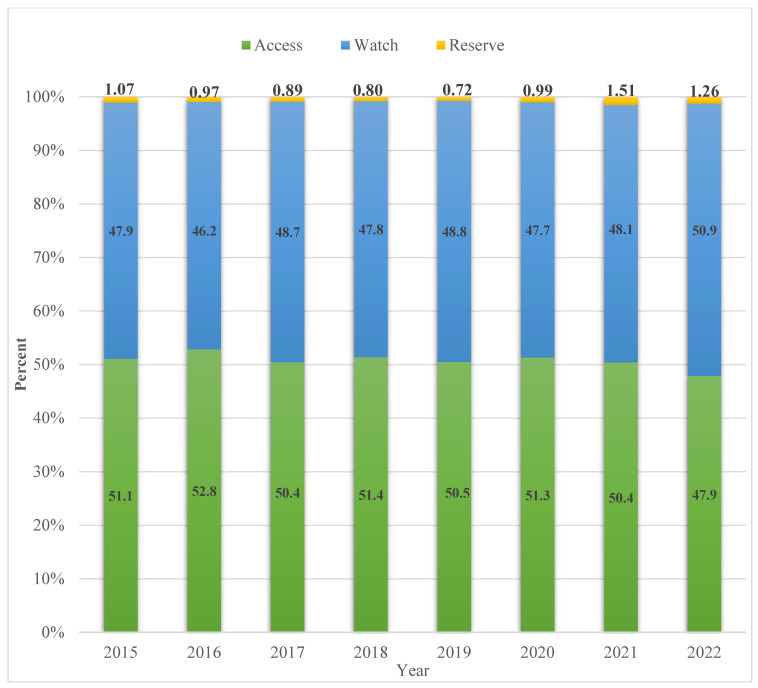
Yearly antibiotic community consumption (packages dispensed) using the WHO AWaRe classification, Cyprus, 2015–2022.

**Table 1 antibiotics-13-00052-t001:** Total community antibiotic consumption by district, overall, and per 1000 inhabitants (in packages dispensed), Cyprus, 2015–2022.

				Antibiotic Consumption (Packages Dispensed)				
District	Pharmacies (All)	Pharmacies (Sampled)	Population	Total	Mean	Standard Deviation	95% CI	Mean Yearly per 1000 Inhabitants
Nicosia	188	77	343,786	1,938,852	242,357	45,121	[204,634; 280,079]	705
Limassol	173	77	253,316	1,692,360	211,545	20,799	[194,157; 228,933]	835
Larnaca	78	53	153,799	937,579	117,197	9843	[108,968;125,427]	762
Paphos	69	30	99,517	706,475	88,309	12,860	[77,558; 99,060]	887
Famagusta	39	19	54,282	420,779	52,597	9104	[44,986; 60,209]	969
Total	547	256	904,700	5,696,045	712,005	95,181	[632,432; 791,578]	787

**Table 2 antibiotics-13-00052-t002:** Outpatient antibiotic use per year expressed as number of orally administered packages per 1000 inhabitants per year, by district and overall, Cyprus, 2015–2022 (*n* = 5,696,045 packages).

	Nicosia	Limassol	Larnaca	Paphos	Famagusta	Cyprus
2015	577	750	717	833	812	691
2016	586	758	669	772	722	677
2017	620	840	762	822	858	742
2018	581	739	714	709	888	681
2019	763	868	767	909	1086	828
2020	833	916	841	990	1158	894
2021	765	845	767	956	1096	829
2022	916	966	859	1108	1132	954
Difference % 2015–2022	58.80%	28.80%	19.80%	33.00%	39.40%	38.10%

**Table 3 antibiotics-13-00052-t003:** Community antibiotic consumption (packages dispensed) using ATC classification, yearly per 1000 inhabitants, Cyprus 2015–2022.

	Tetracyclines (J01A)	Penicillins (J01C)	Cephalosporins and Other Beta lactam Antibacterials (J01D)	Sulfonamides and Trimethoprim (J01E)	Macrolides, Lincosamides and Streptogramins (J01F)	Quinolones (J01M)	Other Groups (J01B, J01G, and J01R)
2015	78	263	136	8	88	106	1
2016	83	261	122	7	86	106	1
2017	80	277	143	8	100	121	1
2018	86	245	125	9	89	114	1
2019	86	306	164	10	122	123	1
2020	100	320	149	17	128	156	2
2021	111	270	135	17	122	148	1
2022	94	322	164	18	170	158	1
% Change 2015–2022	+20.5%	+22.4%	+20.6%	+125.0%	+93.2%	+49.1%	0%

**Table 4 antibiotics-13-00052-t004:** Community antibiotic consumption (packages dispensed) using the AWaRe classification, yearly by district, Cyprus, 2015–2022.

AWaRe	Cyprus			Nicosia			Limassol			Larnaca			Paphos			Famagusta		
	A	W	R	A	W	R	A	W	R	A	W	R	A	W	R	A	W	R
2015	51%	48%	1%	51%	49%	1%	49%	50%	2%	53%	47%	1%	54%	45%	1%	54%	45%	1%
2016	53%	46%	1%	52%	47%	1%	52%	47%	1%	54%	45%	1%	54%	45%	2%	54%	46%	1%
2017	50%	49%	1%	49%	50%	1%	50%	49%	1%	52%	48%	1%	53%	46%	1%	52%	47%	1%
2018	51%	48%	1%	50%	49%	1%	50%	49%	1%	54%	45%	1%	53%	46%	1%	53%	46%	1%
2019	50%	49%	1%	49%	50%	1%	49%	50%	1%	54%	46%	0%	51%	48%	1%	52%	47%	1%
2020	51%	48%	1%	52%	47%	1%	48%	51%	1%	54%	45%	1%	52%	47%	1%	51%	47%	2%
2021	50%	48%	2%	52%	47%	1%	46%	52%	2%	53%	47%	1%	50%	49%	1%	53%	45%	2%
2022	48%	51%	1%	48%	51%	1%	46%	53%	1%	51%	49%	1%	49%	50%	1%	47%	51%	2%
Difference % 2015–2022	−3%	3%	0%	−3%	2%	1%	−3%	3%	0%	−2%	2%	0%	−5%	5%	0%	−6%	6%	0%

A: ‘Access’; W: ‘Watch’; R: ‘Reserve’.

## Data Availability

Additional data are available upon request from corresponding author.

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
