# Peer review of "Community Antibiotic Consumption in Cyprus for the Period 2015 to 2022"

_antibiotics, 2024, doi:10.3390/antibiotics13010052_

Round 1

Reviewer 1 Report

Comments and Suggestions for Authors

This is a good an important manuscript.

My only concern regards the fact that only about 50% of the phrmacies were included. I suggest (if data available) comparing the Pharmacies that were included to those who did not. In terms of location and other parameters. This comparison may support the assumption that the study represents all Cyprus.

Reviewer 2 Report

Comments and Suggestions for Authors

Comments on the Quality of English Language

Round 2

Reviewer 2 Report

Comments and Suggestions for Authors

The revised manuscript is now a much improved and better version. Authors have considered and addressed the comments very critically, sincerely,  diligently and sufficiently. Discussion and Conclusions section are nicely restructured and enriched with additional references. As I can observe, Grammar deficiencies have been rectified and references have also been restyled as per desired format.

In the light of the above, I am delighted to recommend the manuscript for acceptance and publication.